# Molecular Epidemiology and Genetic Diversity of Human Respiratory Syncytial Virus in Sicily during Pre- and Post-COVID-19 Surveillance Seasons

**DOI:** 10.3390/pathogens12091099

**Published:** 2023-08-28

**Authors:** Fabio Tramuto, Carmelo Massimo Maida, Walter Mazzucco, Claudio Costantino, Emanuele Amodio, Giuseppe Sferlazza, Adriana Previti, Palmira Immordino, Francesco Vitale

**Affiliations:** 1Department of Health Promotion, Mother and Child Care, Internal Medicine and Medical Specialties “G. D’Alessandro”—Hygiene Section, University of Palermo, 90134 Palermo, Italy; carmelo.maida@unipa.it (C.M.M.); walter.mazzucco@unipa.it (W.M.); claudio.costantino01@unipa.it (C.C.); emanuele.amodio@unipa.it (E.A.); palmira.immordino@unipa.it (P.I.); francesco.vitale@unipa.it (F.V.); 2Regional Reference Laboratory for Molecular Surveillance of Influenza, Clinical Epidemiology Unit, University Hospital “Paolo Giaccone”, 90127 Palermo, Italy; giuseppe.sferlazza@policlinico.pa.it (G.S.); adriana.previti@policlinico.pa.it (A.P.)

**Keywords:** hRSV-A, hRSV-B, molecular epidemiology, Italy, Sicily, GA2.3.5a, GB5.0.5a, GB5.0.4a, ON1, BA9, genotype

## Abstract

Human respiratory syncytial virus (hRSV) is an important pathogen of acute respiratory tract infection of global significance. In this study, we investigated the molecular epidemiology and the genetic variability of hRSV over seven surveillance seasons between 2015 and 2023 in Sicily, Italy. hRSV subgroups co-circulated through every season, although hRSV-B mostly prevailed. After the considerable reduction in the circulation of hRSV due to the widespread implementation of non-pharmaceutical preventive measures during the COVID-19 pandemic, hRSV rapidly re-emerged at a high intensity in 2022–2023. The G gene was sequenced for genotyping and analysis of deduced amino acids. A total of 128 hRSV-A and 179 hRSV-B G gene sequences were obtained. The phylogenetic analysis revealed that the GA2.3.5a (ON1) and GB5.0.5a (BA9) genotypes were responsible for the hRSV epidemics in Sicily.; only one strain belonged to the genotype GB5.0.4a. No differences were observed in the circulating genotypes during pre- and post-pandemic years. Amino acid sequence alignment revealed the continuous evolution of the G gene, with a combination of amino acid changes specifically appearing in 2022–2023. The predicted N-glycosylation sites were relatively conserved in ON1 and BA9 genotype strains. Our findings augment the understanding and prediction of the seasonal evolution of hRSV at the local level and its implication in the monitoring of novel variants worth considering in better design of candidate vaccines.

## 1. Introduction

Human respiratory syncytial virus (hRSV) is a major cause of lower respiratory tract infections (LRTIs) in infants and young children [1], but it is also responsible for a significant proportion of LRTIs in adults with comorbid conditions and the elderly [2,3].

Based on the antigenic diversity, there are currently two major antigenic groups that are known to circulate in the human population, hRSV-A and hRSV-B, which are distinguished by polyclonal and monoclonal antibodies [4].

It has been reported that variants within each antigenic group may present differential clinical characteristics, such as clinical severity, transmission, and mortality, as well as variations in treatment efficacy [5,6]. However, research comparing the clinical and biological relevance of hRSV genetic diversity has failed to reach a consensus so far. The reason for the contradicting conclusions is likely multi-factorial and attributable to differences in cohort demographics, sample size, and pre-existing immunity [7].

In the light of the genetic variability, found especially in the second hypervariable region (HRV2) of the attachment glycoprotein G gene, both subgroups are classified into several genotypes, which have been historically used for molecular characterization in support of epidemiological assessments on the seasonal epidemics and population dynamics of hRSV [8].

Nonetheless, there is a lack of consensus regarding the criteria to be adopted to allocate genotypes so far [8,9,10,11,12].

Outbreaks of hRSV occur annually and seasonal patterns are observed globally, although with variation from year to year [13]. With regards to weather conditions, in temperate climates, epidemics are characterized by peaks of activity focused during the cold months in the winter, with a fair degree of variability in the timing and duration.

In Sicily, a retrospective cross-sectional study was carried out over five consecutive surveillance seasons, before the emergence of SARS-CoV-2 as a major global concern, reporting an overall hRSV prevalence of 8.1% among subjects with Influenza-like illness (ILI) or severe acute respiratory infection (SARI) [14]. During the season 2020–2021, as consequences of Public Health interventions, social distancing, and movement restrictions settled to counter the pandemic spread of COVID-19 hRSV was not detected in Italy [15,16], confirming the trend described by other authors [17,18,19,20].

Recent studies showed that this prolonged lack of virus exposure resulted in a decline of hRSV-specific immunity [21,22]. Hence, after the gradual relaxation of containment measures occurred in 2021, hRSV circulation has intensified worldwide, with increasing transmission rates in all population groups and an earlier-than-usual start of the season [23,24,25].

The present study was conceptualized to investigate the molecular epidemiology of hRSV in Sicily and aimed to describe the phylogenetic relationships and differences in the genetic characteristics of circulating viral strains throughout seven winter seasons, including the pre- and post-pandemic era.

Therefore, we also evaluated whether hRSV responsible for the most recent outbreak in 2022–2023 had genomic profiles different from viruses identified in the pre-COVID-19 seasons.

## 2. Materials and Methods

### 2.1. Specimen Collection

Oropharyngeal swabs were collected through the running surveillance program in Sicily, within the national network InfluNet, from patients meeting the European case definitions of ILI or SARI [26,27].

More in detail, a case of ILI was defined as a person presenting a sudden and rapid onset of at least one of the following systemic symptoms: fever or feverishness, malaise, headache, myalgia; and at least one of the following respiratory symptoms: cough, sore throat, shortness of breath. A case of SARI was defined as a patient with an acute respiratory infection that required hospitalization.

Samples covered seven winter seasons (October to April) between 2015–2016 and 2022–2023, excluding the 2020–2021 because of the disruption of the national surveillance system in this season caused by the strong impact of the COVID-19 pandemic. All specimens were conferred to the Sicilian Regional Reference Laboratory (RRL) of the InfluNet network, operating at the University Hospital “AOUP P. Giaccone” of Palermo (Palermo, Italy), and stored at −80 °C until further use.

### 2.2. Laboratory Methods

Viral RNA was extracted from specimens using a QIAamp Viral RNA extraction kit (QIAGEN) and tested for hRSV-A and hRSV-B by means of singleplex one-step real-time (rt) retro-transcription assays [14].

Each hRSV positive sample with a rt-PCR cycle threshold (Ct) value ≤ 30 was considered suitable for DNA sequencing of near full-length hRSV G gene.

To this purpose, two different sets of oligonucleotides specific for hRSV-A or hRSV-B were used (oligonucleotide sequences available upon request). Briefly, for each hRSV subgroup, partially overlapping DNA fragments were amplified by using the Superscript III One-Step RT-PCR System with Platinum Taq DNA Polymerase (Invitrogen, Carlsbad, CA, USA).

Successfully amplified PCR products were purified using the Amicon Ultra Centrifugal Unit (Merck, Darmstadt, Germany) according to the manufacturer’s instructions. Then the purified products were sequenced using the BigDye™ Terminator v3.1 Cycle Sequencing Kit (Applied Biosystems, Foster City, CA, USA) and purified with BigDye XTerminator™ Purification Kit (Applied Biosystems, Foster City, CA, USA), before using an ABI Prism 3130xl Genetic Analyzer (Applied Biosystems, Foster City, CA, USA).

### 2.3. Phylogenetic Analysis

To place our datasets into a global context, viral sequences obtained in this study were aligned with reference viral strains of the main hRSV-A and hRSV-B genotypes obtained from GenBank. Sequences without geographic association or sampling date were excluded.

For each hRSV subgroup, a multiple G gene nucleotide sequences alignment was obtained by using a Clustal Omega tool at EMBL-EBI (https://www.ebi.ac.uk/Tools/msa/clustalo accessed on 27 August 2023) and then inspected to identify substitutions or insertions/deletions and manually edited with CLC Genomic Workbench version 23.0.1 (QIAGEN Insight) before investigating the evolutionary relationships.

Preliminary analyses were performed by using Netxclade tool v2.14.1 available online (https://clades.nextstrain.org accessed on 27 August 2023), to assign the clade to each sequence [10] and to explore the most probable phylogenetic placement.

Phylogenetic trees of hRSV-A and hRSV-B were constructed using the Neighbor-Joining method implemented in the MEGA X package [28]. A bootstrap re-sampling analysis was performed (1000 replicates) to test the robustness of tree topology. The best-fit evolutionary model and parameters were calculated and the Tamura-Nei model of nucleotide substitution (TN93 + G + I) was estimated as the most appropriate for the datasets (sequence datasets are available upon request).

### 2.4. Analysis of Deduced Amino Acid Sequences and Mutations

Deduced amino acid (AA) sequences were predicted for both subgroups with standard genetic code and were compared to their respective reference sequences (GenBank accession number: JN257693 for hRSV-A and AY333364 for hRSV-B). Identified AA sequences that differed from the prototype strains were defined as mutants.

### 2.5. Entropy Analysis and N-Linked Glycosylation Sites

Shannon entropy analysis was carried out in BioEdit v7.7 [29] to evaluate the heterogeneity in amino acid positions of the second hypervariable region of the G gene on aligned AA datasets of hRSV-A and hRSV-B Sicilian strains. A threshold value of 0.2 was set to define conserved (<0.2) or variable (>0.2) sites.

The potential N-glycosylation (Asn-X-Ser/Thr) sites were predicted by using the NetNGlyc 1.0 [30].

## 3. Results

This study explored hRSV molecular epidemiology and variability in Sicily from 2015 to 2023. A total of 16,172 oropharyngeal swabs were collected and preliminarily tested for hRSV. Over the whole period, a prevalence of 6.4% (n = 1039/16,172) was found (range: 5.9–11.9%), considering the disruption of the national ILI surveillance system occurred in 2020–2021 due to the strong impact of COVID-19 pandemic (Figure 1).

The hRSV-A accounted for 43.0% (n = 447/1039) of positive samples, while hRSV-B was reported in 54.5% (n = 566/1039). In 2.5% (n = 26/1039), hRSV-A and hRSV-B were co-detected. Although hRSV subgroups co-circulated during the study period, hRSV-B mostly remained the dominant one.

Overall, 307 G protein ectodomain gene sequences were successfully obtained (128 hRSV-A and 180 hRSV-B) and their seasonal distribution is presented in Table 1. To investigate the genetic relationship between Sicilian strains and global circulating strains in recent years, we constructed two different genomic datasets including representative GenBank sequences of previously published genotypes, defined according to the classification proposed by Goya S and colleagues [10].

A preliminary phylogenetic analysis showed that hRSV-A sequences entirely belonged to the genotype GA2.3.5a (equivalent to the genotype ON1). Almost all hRSV-B nucleotide sequences were classified as GB5.0.5a, while only one sequence obtained in the first season of the study was classified as GB5.0.4a (both equivalent to BA strains).

Phylogenetic reconstructions of simplified datasets were depicted in Figure 2 and Figure 3. Apart from some exceptions, sequences clustered into several distinct clades which mirror variability and evolution experienced during the different epidemic seasons, suggesting multiple introductions of both hRSV-A and hRSV-B in Sicily.

It is worth noting that after the significant reduction in the circulation of hRSV, that took place with the emergence of COVID-19 in 2020–2021, viral strains detected during the two most recent seasons (2021–2022 and 2022–2023) belonged to the same genotypes identified in the pre-COVID-19 era, although the G gene sequences homogeneously grouped in separated clusters.

Numerous amino acid changes occurred with variable frequencies over the hRSV seasonal outbreaks considered in the present study (Figure 4, Appendix A).

Changes at the amino acid positions are shown by lines and the length of each line is proportional to the relative frequency (%) of the AA substitution within the dataset of the study sequences (only mutations at frequency ≥2% are depicted).

The sequence corresponds to the 210–322 amino acids of the ON1 prototype strain ON67-1210A (GenBank accession number: JN257693) for hRSV-A and the BA9 prototype strain BA4128/99B (GenBank accession number: AY333364) for hRSV-B.

The boxes with gray shading and dotted lines indicate the analogous AA region followed by the duplicated AA region (23-amino acid for hRSV-A and 20-amino acid for hRSV-B, respectively).

Compared with the ON1 prototype, all hRSV-A strains had conserved stop codons at position 322 of the G gene and exhibited several AA substitutions, especially in HVR2.

The most common non-synonymous mutations were L298P/S (77.3%), L274P (58.1%), and Y304H (57.0%), whereas I243S, E262K, and T320A/I/K were identified in about one-third of the strains (Figure 4A).

Of note, a set composed of fifteen mutations (T80M, L142S, P146S, E224V, P234L, L247P, L274P, G296S, L298P, Y304H, L310P, S311P, S314P, S317F, and T320A) marked the last season 2022–2023, some of which emerged for the first time (underlined in the above list) (Appendix A).

In total, the Sicilian hRSV-B strains showed a panel of 136 unique AA substitutions interspersed within the G gene, when compared with the BA9 prototype. Several AA substitutions were particularly conserved, being represented at very high frequency (Figure 4B, Appendix A).

All sequences exhibited the stop codon at position 320 of the gene, as well as at positions 157 and 159 caused by a 6-nucleotide deletion (Appendix A).

Five more AA changes (L105P, T138S, I200T, K218P/T, and L219P) were reported in 100% of strains, while a few other (T107A/D, Y112H/Q, R136I/T, I281T, S247P, *316Q, and V271A) exceeded 90% of frequency (underlined mutations are included in the 2nd HVR of the G gene) (Figure 4B, Appendix A).

Similarly, to what was observed for hRSV-A, some AA variations strictly correlated with specific epidemic seasons. This is the case for T130I, frequently reported in 2015–2016 and then definitely lost in the following seasons, as well as A131T exclusively documented between 2017–2018 and 2021–2022. Moreover, the mutation T254I, located in the 60-nucleotide duplicated region or the two AA substitutions H287Y/K314R, were maintained until the season 2019–2020 and 2021–2022, respectively.

Some mutations, such as T290I, became dominant in 2017–2018 and, finally, a panel of four new mutations (S100G, P216S, K258N, S277P) emerged with the last surveillance season 2022–2023, mimicking the trend also observed in hRSV-A (Appendix A).

Shannon entropy analysis of HRV2 of the G gene was fulfilled for all hRSV-A and hRSV-B Sicilian strains (Figure 5). Entropy values varied in the range 0–0.74 for hRSV-A and 0–0.93 for hRSV-B, with a threshold value equal to 0.2.

The analysis of hRSV-A sequences identified 33 different variable sites distributed throughout the gene. Eleven of these sites (position 224, 243, 248, 258, 262, 274, 298, 303, 304, 310, and 320) had entropy values greater than 0.5 (underlined sites were included in the 23 AA analogous and duplicated regions), while the AA position 320 in the C-terminal end of the G protein showed the highest variability with an entropy of value 0.74.

A total of 22 variable sites were detected in hRSV-B, seven of which (position 223, 227, 254, 270, 287, 290, and 312) with entropy value > 0.5 (underlined sites were present in the 20 AA analogous and duplicated regions). The AA with the highest variability (entropy value = 0.93) was at position 223, located in the first section of the second hypervariable region.

Table 2 reports the predicted N-linked glycosylation sites in the G gene of Sicilian hRSV-A and hRSV-B. Five sites were predicted in hRSV-A. The first one, observed in all strains, was located at position 85, whereas two sites predicted at position 103 and 237 were found in the majority (99.2%) of hRSV-A included in the study. A fourth site was located at position 135 (95.3%), while 66.4% of strains showed a potential glycosylation site at position 318, not present in hRSV-A strains collected in 2022–2023.

All hRSV-B had two predicted sites at AA positions 81 and 86, while two other sites were specified at positions 296 and 310 of the second hypervariable region of the G gene in 98.3% and 67.8%, respectively. The latter one was progressively lost since 2017–2018 and is no longer observed among hRSV-B strains identified from the season 2019–2020 onwards, being linked to the AA substitution T312I.

Finally, K258N led to the acquisition of a novel N-linked glycosylation site in almost all strains circulating in the last season 2022–2023 (78.8%), and already occasionally observed in 2021–2022.

## 4. Discussion

In the pre-pandemic years, hRSV circulated in Sicily with epidemics typically starting in October-November up to the first quarter of the following year [14]. The emergence of the COVID-19 pandemic caused a significant decline in several respiratory infectious diseases in the community [31,32], including hRSV infections [33,34], because of the implementation of non-pharmaceutical preventive measures adopted to counteract the spread of SARS-CoV-2.

In addition, prolonged low exposure to hRSV during the season 2020–2021 may have favored a sharp decrease of protective immunity at the population level [21,25], which may have represented a major factor for the resurgence of the virus in 2021–2022.

Herein, we report the co-circulation of hRSV-A and hRSV-B in Sicily over seven surveillance seasons. Overall, seasonal variations were observed in the relative proportions, with hRSV-B prevailing over the other subgroups, except for the 2019–2020 season, depicting specific local differences in the epidemiology of hRSV in comparison with other countries [18,35].

The phylogenetic analyses of the nucleotide sequences included in the present study outlined a single genotype for hRSV-A classified as GA2.3.5a, while almost all hRSV-B strains belonged to the genotype GB5.0.5a, with the exception of a single strain collected in the season 2015–2016 and classified as GB5.0.4a (formerly named ON1 and BA9, respectively).

The phylogenetic trees highlighted the similarity between the viral genotypes responsible for the winter outbreaks in the pre-pandemic years and those driving the seasons 2021–2022 and 2022–2023, suggesting that the abrupt reappearance of hRSV in Sicily may be due to the diminished immunity in the general population, rather than to the surge of novel hRSV variants, as also affirmed by other authors [18,25].

Nonetheless, it remains to be determined whether the distinct panel of mutations that emerged during the last season in both subgroups may have simply provided evidence of common evolutionary changes or may have had a role in the unexpected increase of hRSV cases in 2022–2023.

Altogether, the phylogenetic analyses suggest a multiple introduction of hRSV in Sicily as depicted by independent clusters of G gene sequences with dominant lineages varying over time, consistent in terms of surveillance season and, often, of place of origin.

These dominant lineages contained shared AA changes differentiating them from the originally reported genotypes.

More in detail, the Sicilian genetic sequences showed a total of 116 and 138 different AA substitutions in the G gene of hRSV-A and hRSV-B, when compared with the representative strains.

Among ON1 strains, K134I, P146L/S/T/I, I243S, E262K, L274P, L298P/S, Y304H, L310P, L314P, and T320A/I/K were the top 10 most frequently detected AA mutations, which were also described in other European countries [36,37,38,39,40] and elsewhere [41,42,43,44,45,46].

However, it is worth noting that the novel combination of AA mutations, which marked the most recent season 2022–2023, emerged in Sicily for the first time: T80M, E224V, P234L, L247P, G296S, and S311P. To the best of our knowledge, these six signature AA changes have not been described before, except for the single AA substitution G296S that was found in a couple of South Korean genomes collected in 2014–2015 [47] and the three mutations L274P, G296S, and S311P identified in a cluster of Chinese ON-1 strains circulating in late 2021, after a peak of COVID-19 [48]. Nevertheless, it should be considered that very few papers concerning the analysis of genomes of viruses isolated during the 2022–2023 season have been published yet. Thus, the forthcoming reports may help to corroborate the potential role of these panels of mutations in promoting the emergence of viral variants with selective fitness advantage and, thus, global dispersal.

For hRSV-B, about twenty major AA changes were identified in the G gene of most Sicilian BA9 strains (frequency > 70%). Eight of these AA: K218P/T, L219P, L223M/P/S/T, I281T, H287Y, Q313*, K314R, and *316Q were outside the duplicated region of HVR2 and four AA: S247P, T254I, T270A/I, and V271A within the duplicated region.

In South Korea [44], some of these BA9 genotype-specific mutations were found at very high frequency, such as V271A, I281T, and H287Y (range: 70.3–86.5%), and in hRSV strains circulating in the period 2010–2019 most Taiwanese BA9 strains harbored the codon AA changes K218T, L223P, S247P, I281T, and H287Y (outside the duplication site).; T270I and V271A (within the duplication site) [45].

Additionally, a large part of these AA variations has been widely reported throughout the world [8,44,49,50,51]. Further, as similarly observed for hRSV-A, four AA mutations were uniquely shared by hRSV-B strains circulating in Sicily in the season 2022–2023 (S100G, P216G, K258N, S277P), which were also partially documented in Austria one year earlier [18].

Shannon entropy analysis was performed to identify the variable sites on the G gene of hRSV-A and hRSV-B. Among the thirty-three variable sites detected in the ON-1 genotype, five were found in the duplicated region. The site at position 274 was also reported in Saudi Arabia alone [52], as well as the co-occurring AA at positions 262, 274, 298, 303, and 304 indicating their high variability in hRSV-A strains collected in the Philippines [53].

With regards to hRSV-B, a total of twenty-two sites were identified in BA9 lineage, and seven different AA were reported to have high entropy values, pointing out variations at these positions. In particular, the AA at position 270, located within the 20AA duplicated region of HRV2, was reported in India [49] as well as in other countries in the Middle East and the Far East [52,53].

Overall, the Shannon entropy analysis showed that hRSV-A had more variable sites than hRSV-B, suggesting a higher potential for substitution in this subgroup and a greater evolutionary selection pressure [53,54].

Finally, N-linked glycosylation of the G protein represents a crucial landmark of viral antigenicity being involved in the expression of epitopes affecting antibody recognition and, thus, immune evasion [55,56,57]. Most of the predicted sites of the hRSV-A strains were highly conserved, as previously observed in Kenya [58], Madagascar [42], and Europe [59], while the AA change in T320A ultimately contributed to a glycosylation loss in the Sicilian strains from the 2022–2023 season.

In the same way, hRSV-B strains here documented had the N-linked glycosylation sites typically conserved at the position 296 and 310 [42,45,53]; this latter one was lost since 2020–2021, while K258N substitution within the 60-nt duplication of HVR2 led to the emergence of a novel site in the most recent circulating viruses, which could have played a role in the increased spread of this strains in the last seasons.

There were some limitations in our study. First, our research was limited to G gene sequences, although whole-genome sequencing would have contributed more information on genetic diversity suitable to better identify genetic signature and/or non-synonymous substitutions, also covering other fragments such as the fusion (F) protein gene, a major target of immunoprophylactic monoclonal antibodies and vaccines [60]; next, this study was confined to a single region of Italy, whilst Sicily may represent a privileged location to capture novel viral variants, due to its position in the middle of the Mediterranean [14,61,62]; last, we were not able to estimate the potential role of subgroups or specific mutations in viral phenotype since no clinical data were available.

This latter aspect is still being debated globally. A different impact of hRSV-A and hRSV-B on clinical outcomes and disease severity has been reported in the most recent literature [63], as well as in the past [64]. Nevertheless, other authors found no association between clinical outcomes and hRSV subgroups or genotypes [65]. Additionally, it cannot exclude a possible causative relationship with individual factors, such as innate and adaptive immune responses [64,66]. Furthermore, it would be interesting to investigate whether and how the mutational patterns found in the last seasons and reported herein, may have had a role in the resurgence of viral circulation, also at a higher level than in the past [63].

In summary, our data seem to suggest that the re-emergence of hRSV in Sicily was more likely due to a significant decline of naturally acquired protective antibody levels in the absence of viral exposure [21,22], also for a short period, in the context of non-pharmaceutical interventions adopted during the COVID-19 pandemic.

Consequently, this raises the need for further reflection on the duration of vaccine-induced immunity with the aim of guiding decision-makers in programming the most effective immunization strategies.

In conclusion, this is the first study exploring the molecular epidemiology of hRSV in Sicily, the largest island in the Mediterranean Sea. Our findings add further data and highlight the need for harmonization and enhanced molecular surveillance of hRSV genotypes. In this context, the implementation of whole-genome sequencing may have an additional value, contributing to improving the knowledge of evolutionary pathways of this virus and its global spread, but also to give insight into mutation profiles worth considering in the analysis of the effectiveness of forthcoming vaccines, as well as in a better design of future preventive measures [67].

## Figures and Tables

**Figure 1 pathogens-12-01099-f001:**
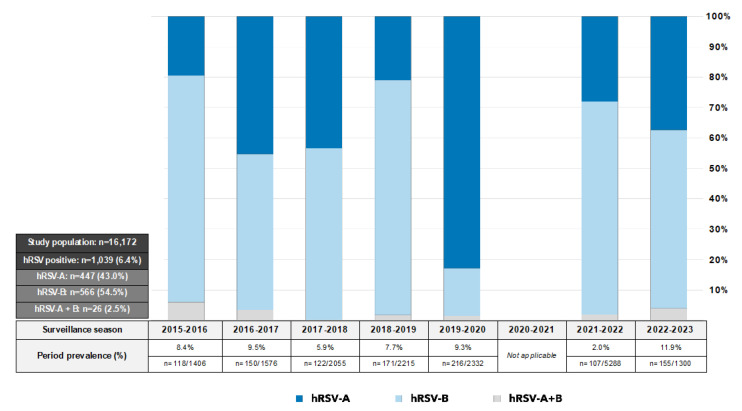
Seasonal distribution of hRSV-A and hRSV-B positive samples identified in Sicily between 2015–2016 and 2022–2023.

**Figure 2 pathogens-12-01099-f002:**
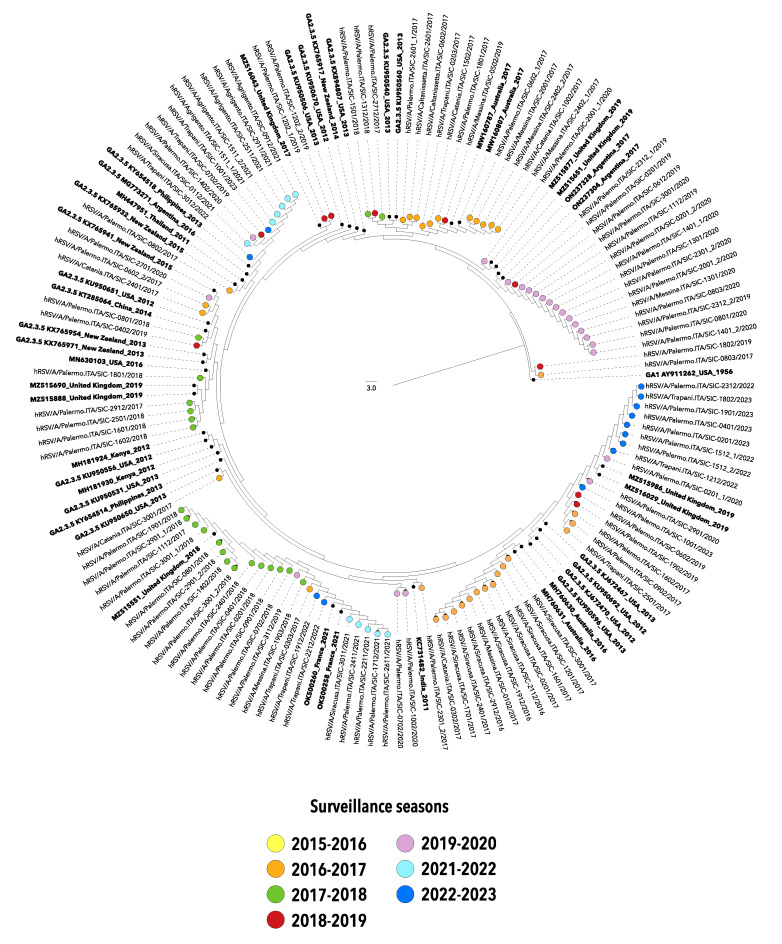
Neighbor-joining (NJ) phylogenetic tree of nucleotide sequences from the second hypervariable region of the G gene of hRSV-A strains collected in Sicily between 2015 and 2023. The study sequences are indicated by solid circles, differently colored according to the surveillance season. Reference sequences are indicated in bold and are expressed in the following format: Genotype/GenBank accession number/country/year. The strain GA1 Y911262_USA_1956 was used as an outgroup. Genotypes are defined according to the classification proposed by Goya S et al. [10].

**Figure 3 pathogens-12-01099-f003:**
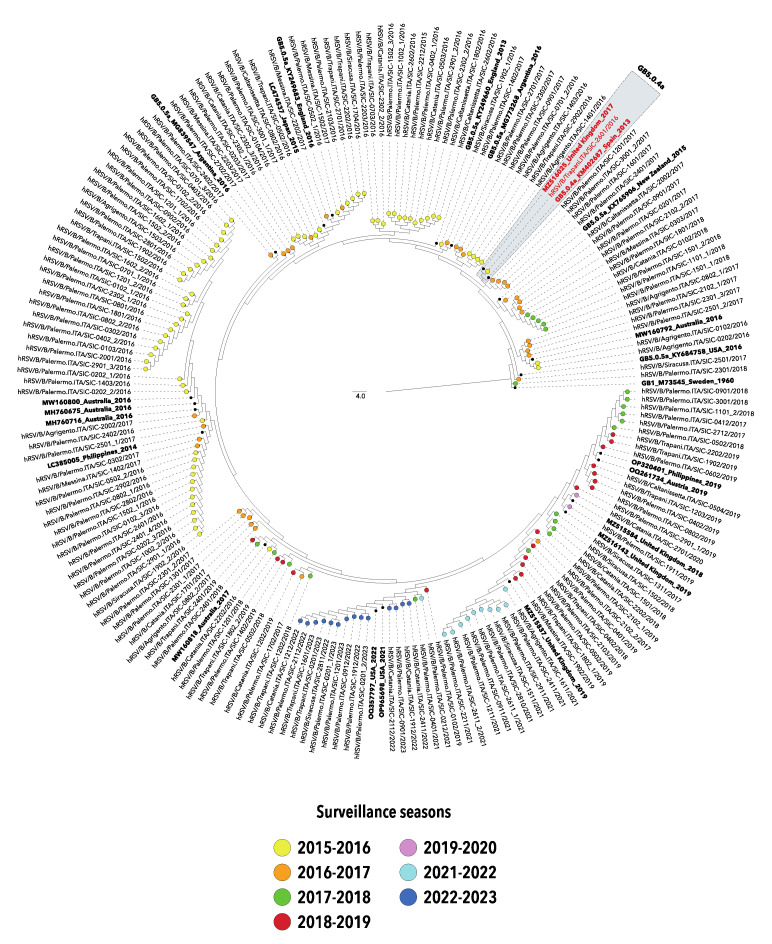
Neighbor-joining (NJ) phylogenetic tree of nucleotide sequences from the second hypervariable region of the G gene of hRSV-B strains collected in Sicily between 2015 and 2023. The study sequences are indicated by solid circles, differently colored according to the surveillance season. The unique strain belonging to GB5.0.4a is indicated red and highlighted in gray. Reference sequences are indicated in bold and are expressed in the following format: Genotype/GenBank accession number/country/year. The strain GB1 M73545_Sweden_1960 was used as an outgroup. Genotypes are defined according to the classification proposed by Goya S et al. [10].

**Figure 4 pathogens-12-01099-f004:**
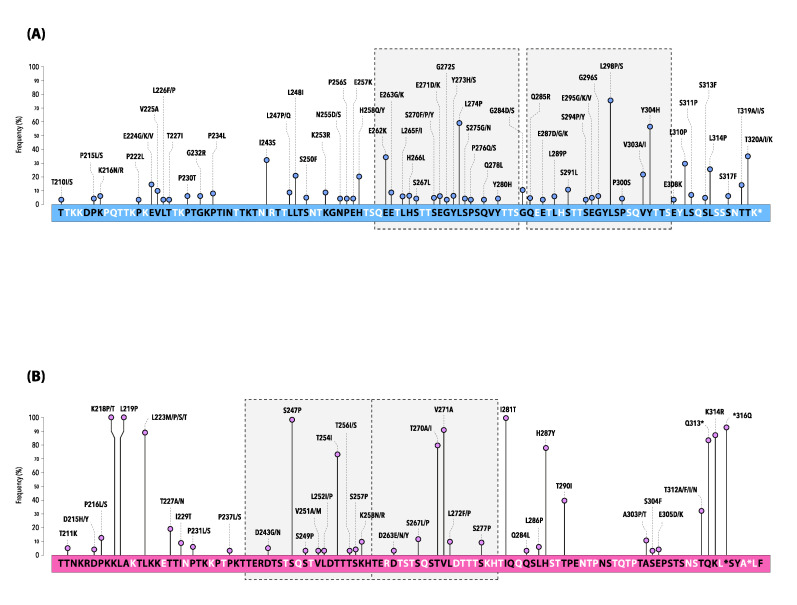
Mutations of the deduced amino acid sequence of the second hypervariable region of the G gene of hRSV-A (**A**) and hRSV-B (**B**) Sicilian strains. * indicates a stop codon.

**Figure 5 pathogens-12-01099-f005:**
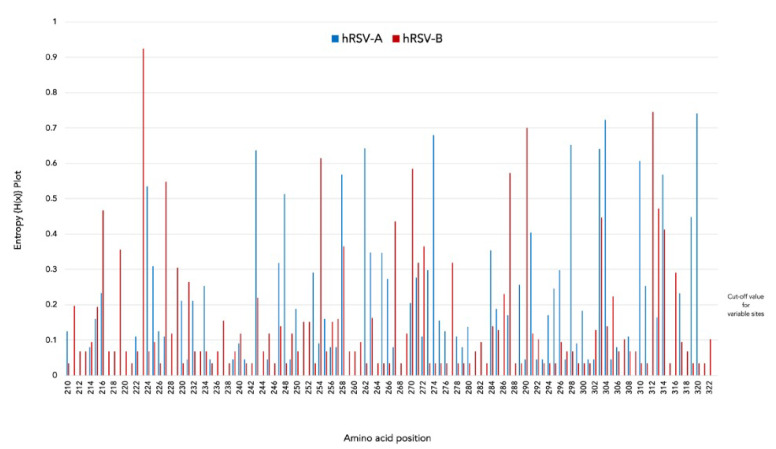
Amino acid variability of the second hypervariable region of the G gene represented by Shannon entropy plot. The threshold value was set at 0.2. Amino acid sites with entropy values < 0.2 are considered conserved, while values > 0.2 are considered variable. Horizontal blue and red bars indicate the analogous AA region followed by the duplicated AA region (23-amino acid for hRSV-A and 20-amino acid for hRSV-B, respectively).

**Table 1 pathogens-12-01099-t001:** Seasonal distribution of hRSV genetic sequences obtained in Sicily. Period: 2015–2023.

	Total Sequences	hRSV-A	hRSV-B
Overall	307	128	179
2015–2016	92	19	73
2016–2017	67	32	35
2017–2018	44	23	21
2018–2019	30	8	22
2019–2020	24	22	2
2020–2021	NA		
2021–2022	23	11	12
2022–2023	27	13	14

NA Not applicable.

**Table 2 pathogens-12-01099-t002:** Predicted N-glycosylation sites in the G gene of hRSV-A and hRSV-B strains identified in Sicily.

	AA Position	Frequency (%)
hRSV-A	N85	100.0
	N103	99.2
	N135	95.3
	N237	99.2
	N318 *	66.4
hRSV-B	N81	100.0
	N86	100.0
	N296	98.3
	N310 **	67.8

* Not reported in the 2022–2023 season. ** Gradually lost since 2017–2018 and no longer reported by 2019–2020.

## Data Availability

Data available upon request.

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
