# Peer review of "Molecular Epidemiology and Genetic Diversity of Human Respiratory Syncytial Virus in Sicily during Pre- and Post-COVID-19 Surveillance Seasons"

_pathogens, 2023, doi:10.3390/pathogens12091099_

Round 1
Reviewer 1 Report
The manuscript entitled “Molecular epidemiology and genetic diversity of respiratory syncytial virus in Sicily during pre- and post-COVID-19 surveillance seasons.” by Tramuto et al. describes the genetic variability in RSV between pre- and post- pandemic RSV infections in Sicily. Authors have analyzed the G gene sequences over seven surveillance seasons in order to determine the cause of increase in RSV cases post COVID-19 pandemic. Interestingly, RSV genotype remained same for the post-pandemic infections which ruled out the possibility of mutations in RSV as a reason for RSV epidemic in Sicily. The N-glycosylation sites were also analyzed. Overall, this is an important study, and the data has been analyzed carefully. However, it requires few changes and need additional information for further improvement.
In abstract and discussion, authors claim that this study will help in designing vaccines. This study only includes G gene data, not the F gene one. The reference authors mentioned about the connection of glycosylation sites to the antigenic determinant does not show anything about RSV. Is it the hypothesis of authors that G gene glycosylation sites can help understanding vaccine binding? If yes, there should be an experimental proof or a reference paper which includes this information.
I understand the limitations of this study to only G gene sequencing. But, if possible, at least F gene should have been sequenced in case whole genome is not possible to sequence. F sequence will give a better understanding of the variations in antigenic sites which can help towards vaccine design.
What is the advantage of analyzing the AA changes in glycoprotein HVR2 region? Can authors explain how this analysis can help improving public health? What is the importance of these mutations?
Lines 281-283- Firstly this paragraph is too introductory to be included here in discussion section and can be a part of introduction. Second, authors referred another paper Goya et al that there is an emergence in novel genotypes which is not mentioned in this paper. Please choose the reference carefully and cite only information that is mentioned in the reference paper.
Discussion looks more like an introduction and a repetition of results. If authors want to explain this much, they can include this in introduction. For examples, lines 294-300 are very descriptive and not suitable for discussion section.
307 G sequences were obtained out of 1039 RSV positive samples. What was the issue with rest of the sequences?
Authors have mentioned the word “genome” in abstract as well as at other places in the manuscript. Only G gene sequences are available, not full genomes. Please change the language accordingly.
Introduction line 32 – it is respiratory syncytial virus.
Line 81- please rewrite this.
Line 96- references strains names should be mentioned.
Lines 311 and 314 are not clear what authors are saying here. Please rephrase for better understanding.
English language looks fine.
Author Response
TO REVIEWERS
We sincerely thank the Reviewers for providing us this opportunity to further revise our manuscript.
A point-by-point description of how each comment was addressed in the manuscript is given below. Original reviewers' comments in boldface, responses in regular typeface.
To Reviewer #1
- In abstract and discussion, authors claim that this study will help in designing vaccines. This study only includes G gene data, not the F gene one. The reference authors mentioned about the connection of glycosylation sites to the antigenic determinant does not show anything about RSV. Is it the hypothesis of authors that G gene glycosylation sites can help understanding vaccine binding? If yes, there should be an experimental proof or a reference paper which includes this information.
Response:
To date, most of hRSV vaccine candidates in the end phase of development, as well as already approved monoclonal antibodies, target the prefusion conformation of the F protein (pre-F) located on the surface of the virion, because of robust neutralising responses in healthy and older adults. Nevertheless, besides the F protein, other hRSV targets have been also considered including the G protein (please see Jenkins VA et al. Vaccines (Basel). 2023 Feb; 11(2): 382).
Therefore, while we totally agree with the Reviewer’s comment, in our opinion, the genetic variation in the G protein gene could help in better design of candidate vaccines.
The reference cited above has been included in the manuscript [59], as well as to more references [52,53] concerning the potential role of hRSV G protein glycosylation sites in immunogenicity.
- I understand the limitations of this study to only G gene sequencing. But, if possible, at least F gene should have been sequenced in case whole genome is not possible to sequence. F sequence will give a better understanding of the variations in antigenic sites which can help towards vaccine design.
Response:
Our team is working on it. In the short term, a new research based on hRSV whole genomes will be proposed. However, we are not able to provide data on the F protein gene so far.
- Lines 281-283- Firstly this paragraph is too introductory to be included here in discussion section and can be a part of introduction. Second, authors referred another paper Goya et al that there is an emergence in novel genotypes which is not mentioned in this paper. Please choose the reference carefully and cite only information that is mentioned in the reference paper.
Response:
We thank the Reviewer for the comment. The phrase reported between line 281 and 283 has been deleted, whereas Goya et al (Emerg. Infect. Dis. 2023, 29, 865-868) has been cited in accordance with the conclusions reported by the authors, which highlight a “diminished protective immunity in the population from low RSV exposure”, as a result of the mitigation measures adopted during the SARS-CoV-2 pandemic.
- Discussion looks more like an introduction and a repetition of results. If authors want to explain this much, they can include this in introduction. For examples, lines 294-300 are very descriptive and not suitable for discussion section.
Response:
We agree with the Reviewer. The paragraph has been removed and the references revised accordingly.
- 307 G sequences were obtained out of 1039 RSV positive samples. What was the issue with rest of the sequences?
Response:
As reported in 2.2 Laboratory methods, only “hRSV positive sample with a rt-PCR cycle threshold (Ct) value ≤30 was considered suitable for genome sequencing of near full-length hRSV protein G gene”.
- Authors have mentioned the word “genome” in abstract as well as at other places in the manuscript. Only G gene sequences are available, not full genomes. Please change the language accordingly.
Response:
The term “genome” has been replaced throughout the manuscript according to the reviewer’s comment.
- Introduction line 32 – it is respiratory syncytial virus.
Response:
We are sorry but the term “syncytial” has been unintentionally missed. The text has been revised.
- Line 81- please rewrite this.
Response:
We sincerely apologize but we do not understand what should be rewritten. Anyway “genome sequencing” has been replace with “DNA sequencing” according to the Reviewer’s comment in point 6.
- Line 96- references strains names should be mentioned.
Response:
Reference strains are indicated in boldface in the Neighbor-joining phylogenetic tree depicted in Figure 2 and 3.
- Lines 311 and 314 are not clear what authors are saying here. Please rephrase for better understanding.
Response:
According to the reviewer’s comment, the paragraph between line 311 and 314 (in the revised version of the manuscript, lines 294-296) has been modified and contextualized with references from Austria [15] and USA [22].
Reviewer 2 Report
The present article evaluates the molecular epidemiology and genetics of the hRSV. This study evaluates the genetic variability from 2015 to 2023. One of the most important flaws is that no data correlates the genotype circulant with the hospitalization cases or disease severity. This is relevant since after SARS-CoV-2 pandemic, the incidence of hRSV cases, hospitalization, and deaths worldwide. The authors must mention and analyze how was the incidence in Italy and discuss it.
Mayor concerns
- The authors must specify the origin of the samples. It is relevant to know the patient's age, disease severity, gender, and other clinical parameters. The author must evaluate the genotype considering this.
- The authors must include information about the genotype and their clinical characteristics in the introduction.
- Besides the author cited the exclusion criteria of the samples, it is relevant that this point be detailed in the manuscript.
- Figures 2,3, and 4 are very small and cannot see.
- The authors must intensely discuss their findings' implications regarding the hRSV infection's clinical parameters.
Minor concern
- Add Human to the title.
- Line 12: Human….
- Grammar must be checked.
Author Response
TO REVIEWERS
We sincerely thank the Reviewers for providing us this opportunity to further revise our manuscript.
A point-by-point description of how each comment was addressed in the manuscript is given below. Original reviewers' comments in boldface, responses in regular typeface.
To Reviewer #2
- The authors must specify the origin of the samples. It is relevant to know the patient's age, disease severity, gender, and other clinical parameters. The author must evaluate the genotype considering this.
Response:
We agree with the Reviewer that the evaluation of possible correlation between genotypes and relevant individual data should be considered. However, as stated at lines 369-370 (section Discussion), no clinical data were available.
- The authors must include information about the genotype and their clinical characteristics in the introduction.
Response:
As reported above (point 1), unfortunately no clinical characteristics were known.
- Besides the author cited the exclusion criteria of the samples, it is relevant that this point be detailed in the manuscript.
Response:
According to the reviewer’s comment, the phrase “from patients meeting the European case definitions of ILI or SARI [11,23]” has been included in the manuscript and a new reference regarding the ECDC ILI and SARI definition has been added.
- Figures 2, 3, and 4 are very small and cannot see.
Response:
We understand the reviewer’s concern. We provided the original figures at full-resolution and we are confident that the editor, in case of acceptance of the paper, will publish the figures as large as possible according to the editorial style of the journal.
- The authors must intensely discuss their findings' implications regarding the hRSV infection's clinical parameters.
Response:
As commented above (points 1 and 2), based on the available data, we are not able to discuss the potential role of our findings in relation to clinical parameters.
- Add Human to the title.
Response:
According to the reviewer’s suggestion, the word “human” has been added to the title of the manuscript.
- Line 12: Human….
Response:
As above, the term “human” has been added in the abstract.
Round 2
Reviewer 1 Report
This is a revised version of the manuscript entitled “Molecular epidemiology and genetic diversity of human respiratory syncytial virus in Sicily during pre- and post-COVID-19 surveillance seasons.” by Tramuto et al. Authors have responded very well for all the comments/concerns point-wise and revised the manuscript accordingly.
Not sure what does “hRSV-B protein G gene” mean. Can it be hRSV G gene in the entire manuscript?
N/A
Author Response
To Reviewer #1
We sincerely thank the Reviewer for all valuable comments.
According to the suggestion, the term “… protein G gene” has been replaced by “… G gene” throughout the manuscript.
Reviewer 2 Report
The authors argue that no clinical data is available to correlate their findings with the patient's age, disease severity, gender, and other clinical parameters. However, the authors did not improve the manuscript as was required, which is necessary to accept the article.
- The authors must include information about the genotype and their clinical characteristics (from the literature) in the introduction.
- Besides the author citing the exclusion criteria of the samples, it is relevant that this point be detailed in the manuscript, not only add more references.
- The authors need to intensely discuss their findings' implications regarding the hRSV infection's clinical parameters described in the literature.
- The authors must mention the study's limitations and speculate the clinical relevance of their findings.
The authors must recheck the grammar.
Author Response
To Reviewer #2
We sincerely thank the Reviewer for all valuable comments. A point-by-point description was addressed in the manuscript hoping to have improved the paper and satisfactorily answered according to the suggestions. All revisions have been highlighted in yellow.
- The authors must include information about the genotype and their clinical characteristics (from the literature) in the introduction.
Response:
As suggested, the introduction has been revised, and a new sentence regarding the potential role of hRSV subgroups in clinical outcomes has been added. References have been modified accordingly.
- Besides the author citing the exclusion criteria of the samples, it is relevant that this point be detailed in the manuscript, not only add more references.
Response:
We agree with the Reviewer. In the section “Materials and Methods”, inclusion and exclusion criteria have been described in more detail and a new reference has been added.
- The authors need to intensely discuss their findings' implications regarding the hRSV infection's clinical parameters described in the literature.
Response:
According to the Reviewer’s suggestion, a new sentence concerning the potential implication of clinical/virological characteristics on hRSV infection and disease severity has been added in the section “Discussion”, as well as some valuable references.
- The authors must mention the study's limitations and speculate the clinical relevance of their findings.
Response:
We sincerely appreciate the Reviewer’s observation. However, in our opinion, the main limitations of our study have been reported, including (a) the lack of genetic sequences other than G gene, (b) the need to extend the analysis to genetic fragment more specifically relevant to vaccine design, such as the F gene, (c) the potential bias related to the epidemiology of a single geographic area of the country and, last but not least, (d) the lack of clinical data which, undoubtedly, would have allowed to speculate on the “clinical relevance” of our findings.
Round 3
Reviewer 2 Report
Thanks to the authors for including the changes suggested by this Reviewer.
I suggest rechecking the grammar once more.